# Prevention and Treatment of Retinal Vein Occlusion: The Role of Diet—A Review

**DOI:** 10.3390/nu15143237

**Published:** 2023-07-21

**Authors:** Maja Lendzioszek, Małgorzata Mrugacz, Anna Bryl, Ewa Poppe, Katarzyna Zorena

**Affiliations:** 1Department of Ophthalmology, Voivodship Hospital in Lomza, 18-400 Lomza, Poland; lendzioszek.majka@gmail.com (M.L.); epoppe@interia.pl (E.P.); 2Department of Ophthalmology and Eye Rehabilitation, Medical University of Bialystok, 15-089 Bialystok, Poland; malgorzata.mrugacz@umb.edu.pl; 3Department of Immunobiology and Environmental Microbiology, Medical University of Gdansk, 80-211 Gdansk, Poland; katarzyna.zorena@gumed.edu.pl

**Keywords:** eye, retinal vein occlusion, diet, lifestyle

## Abstract

Retinal vein occlusion (RVO) is the second most common retinal disorder. In comparison to diabetic retinopathy or age-related macular degeneration, RVO is usually an unexpected event that carries a greater psychological impact. There is strong evidence to suggest that cardiovascular diseases are the most common risk factors in this pathology and it has long been known that a higher consumption of fish, nuts, fruits, and vegetables has a protective effect against these types of conditions. In the last several years, interest in plant-based diets has grown in both the general population and in the scientific community, to the point to which it has become one of the main dietary patterns adopted in Western countries. The aim of this review is to investigate the potential impact of macro- and micronutrients on retinal vein occlusion.

## 1. Introduction

Retinal vein occlusion (RVO) ranks as the second most prevalent retinal disorder, second to diabetic retinopathy, with an estimated incidence of 0.3% to 2.3% depending on ethnicity, and can lead to blindness of a vascular origin [1]. The retinal vasculature gives a non-invasive in vivo insight into the state of the human microcirculation [2]. It is believed that the pathogenesis of retinal vein occlusion is caused by mechanical compression by atherosclerosis of the retinal artery. The central retinal artery and vein share a common adventitious sheath. Changes in the arterial wall associated with atherosclerosis or hypertension increase its stiffness which translates into increased impact on the central retinal vein, leading to turbulent blood flow in its lumen. This cascade of events leads to endothelial damage and ultimately leads to hypercoagulability and occlusion of the vessel [3]. Depending on the anatomical location of the thrombus, the following types are distinguished: central retinal vein occlusion (CRVO), superior or inferior retinal vein occlusion (hemi-CRVO), and branch retinal vein occlusion (BRVO). Typical risk factors for RVO include advanced age, hypertension, diabetes, hyperlipidemia, and glaucoma [4]. In the younger population, special attention is paid to lipid disorders of the blood as the main risk factor [5].

There is ample evidence that cytokines and chemokines contained in the vitreous body are correlated with the occurrence of RVO, especially the interleukin family, VEGF, MMP, LPA-ATX and PDGF [4]. The interleukin family is pro-inflammatory, causing ischemia and macular edema secondary to RVO, and the most important in this disease include IL-6, IL-8, IL-17 and IL-18, which trigger STAT3, MAPK, NF-κB, VEGF pathways and provoke ROS. VEGF inhibits occludin by damaging the basement membrane of endothelial cells and activates MMP-9 to destroy blood–retinal barriers (BRB) and induces ICAM-1 causing leukocyte stasis. VEGF also works by activating the NOX1 and NOX4 proteins, which are dominant in ROS production in RVO. MMP-2 and MMP-7 are involved in the migration of vascular endothelial cells. The LPA-ATX signaling pathway may mediate inflammation in RVO as it activates IL-6, IL-8, VEGF and MMP-9. PDGF-A potentiates VEGF to induce neovascularization. An interesting study was presented by Takuma Neo et al., who used the rabbit retinal vein occlusion model in order to analyze ischemia-induced changes in gene expression profiles [6]. The study revealed that angiogenic regulators Dcn and Mmp1 and the pro-inflammatory factors Mmp12 and Cxcl13 were significantly elevated in RVO retinas. In total, they analyzed 387 genes with more than a 2-fold difference between RVO and controls (upregulated: 333 genes, downregulated: 54 genes). What is more, they confirmed that JAK-STAT, TNFα and NF-κB pathways likely contribute to rabbit RVO pathology and potentially human retinal ischemic disease.

Given the intricate nature of risk factors and their interplay in RVO, it is crucial to identify the most important predictors of the disease early, as well as to introduce appropriate prophylaxis. A well-known part of the Hippocratic oath states–prevention is better than cure. This holds particular significance in thrombotic disorders affecting the venous vascular system of the eye, as they can result in abrupt and painless vision deterioration or visual field abnormalities, potentially leading to complete blindness in severe cases [7]. The main complications of RVO are macular edema, neovascular glaucoma, and hemorrhage into the vitreous chamber [7]. The treatment of complications associated with RVO consists primarily of the intravitreal administration of anti-VEGF preparations in the event of macular edema and photocoagulation of the entire retina in cases of iris neovascularization [8,9]. It has been proven that during RVO, vascular endothelial growth factor (VEGF) is the main cytokine inducing ischemia and neovascularization; therefore, intravitreal anti-VEGF in the event of macular edema following retinal vein occlusion is the first line of treatment [1]. Aflibercept (EYLEA) is a fusion protein that binds VEGF-A, VEGF-B and placental growth factor (PIGF) with a greater affinity than the body’s native receptors. Ranibizumab (Lucentis) is a recombinant humanized IgG1 monoclonal antibody fragment that binds to and inhibits only VEGF-A. Bevacizumab (Avastin) is a humanized antibody that binds all subtypes of VEGF-A. The recommended dose for aflibercept is 2 mg (0.05 mL), for bevacizumab it is 1.25 mg (0.05 mL) and for ranibizumab it is 0.5 mg (0.05 mL) administered by intravitreal injection once every 4 weeks. After the first several injections, some patients continue monthly treatment, some patient are treated at increasing intervals and some patients are checked monthly and treated as needed. Faricimab (Vabysmo) is a promising bispecific drug targeting VEGF-A and the Ang-Tie/pathway [10]. It is a combined-mechanism medication with simultaneous and independent binding on both VEGF-A and angiopoietin-2 (Ang-2). It is believed to have a more lasting effect than previous anti-VEGF drugs in clinical trials. The FDA approved Vabysmo for the treatment of diabetic macular edema and neovascular age-related macular degeneration on January 2022. Another option to deliver effective anti-VEGF doses over a longer period of time is to use a slow-release intraocular device, such as a PDS device [10]. Patients who do not respond to anti-VEGF preparations are recommended to implement intravitreal steroid injection–triamcinolone or dexamethasone implant (DEX, Ozurdex) which reduce pathologically increased capillary permeability and inhibit the expression of cytokines and chemokines. Another method of treating macular edema is focal laser or grid laser which has now lost its importance because of intravitreal drug injections. In patients with ischemic RVO, panretinal laser photocoagulation (PRP) is recommended for treatment of secondary neovascular complications. The use of systemic recombinant tissue plasminogen activator (rtPA), radial optic neurotomy, chorioretinal anastomosis and arteriovenous sheathotomy are extremely rarely used methods due to possible complications such as sudden hemorrhage, visual-field defects or retinal detachment [1]. Pars plana vitrectomy is considered in the presence of attached posterior hyaloids accompanied by persistent macular edema in CRVO. The recommended treatment methods are, unfortunately, associated with regular, lengthy visits to ophthalmological treatment facilities, which for people who are professionally active means exclusion from the labor market, and for the elderly and dependent people, family involvement in the treatment process. These and other disturbances of the quality of life of patients with RVO, such as fear of the second eye being affected by the disease process, fear of not improving after injections, and fear of the future, were demonstrated in the work of Prem Senthil et al. [11].

In light of these facts, is possible to prevent retinal vein occlusion? Is it possible for dietary and lifestyle choices to exert a preventive influence on RVO?

## 2. Plant-Based Diets Reducing the Main RVO Risk Factors: Hypertension, Hyperlipidemia, and Diabetes

The essential role of food ingredients in human physiology has been widely recognized for a considerable period of time. However, it has only recently been recognized that many micronutrients have a profound influence on human health and disease risk. Several dietary factors possess the remarkable capability of modifying the expression of regulatory genes involved in human metabolism [12]. These genes govern crucial processes like cell growth, programmed cell death, and immune system response. Effective control of these processes is critical for preventing various human ailments, including cancer, autoimmune disorders, and inflammatory conditions [13].

The most widely accepted dietary patterns in Western countries include the plant-based diet (PBD), the Mediterranean diet (MD), the Paleolithic diet, low-carbohydrate diets, and low-fat diets. In particular, the plant-based diet is growing in popularity as the healthiest diet. The most important concern with this nutritional approach is the risk of developing nutritional deficiencies in protein, omega-3 fatty acids, vitamin B12, iron, zinc, iodine, vitamin D, and calcium. However, as a diet rich in fruits and vegetables, the PBD is high in fiber, omega-6 fatty acids, antioxidants, and phytochemicals [14]. Various types of diets can be distinguished among the PBD. Vegetarians can be categorized into different groups based on their dietary preferences. Lacto-ovo vegetarians abstain from consuming meat but include dairy products, eggs, and other animal-derived foods in their diet. Lacto-vegetarians, on the other hand, exclude eggs while still consuming dairy products. Ovo-vegetarians exclude dairy products but include eggs as part of their dietary choices. The vegan model excludes all foods of animal origin.

Patients following a plant-based diet tended to have lower blood pressure (BP), body mass index (BMI), and lower total cholesterol (TC), low-density lipoprotein cholesterol (LDL-C), triglycerides (TG), and blood glucose levels compared to omnivorous people [15] (Figure 1).

High blood pressure (BP) is recognized as a significant modifiable cardiovascular risk factor [16]. Dietary patterns play a crucial role in both preventing and managing this condition. By adopting appropriate dietary choices, it is possible to delay the onset of high blood pressure and mitigate its effects. It should be characterized by a low intake of saturated fatty acids, a high intake of fruits and vegetables, limiting the amount of foods with high salt content, and limiting the consumption of alcohol. A meta-analysis of 32 observational studies with 604 participants found an association between vegetarian diets and average reductions in systolic and diastolic blood pressure of 6.9 mmHg and 4.7 mmHg, respectively [17].

Epidemiological research conducted in Western countries has revealed a notable occurrence of hypercholesterolemia, accompanied by a high frequency of cardiovascular disease and associated mortality [18]. Clinical trials have further demonstrated that even a modest reduction of 1% in LDL cholesterol levels can lead to a corresponding decrease of approximately 1% in the risk of major cardiac events, such as heart attacks and strokes [19]. Lifestyle changes, particularly diet and exercise, have been shown to lower LDL levels by up to 30–40% in people at risk for cardiovascular disease, which may play a significant role in preventing and possibly treating this group of diseases [20]. Animal fats such as meat, butter, whole dairy products, as well as tropical coconut and palm oils are typically high in saturated fatty acids (SFAs). In contrast, vegetable fats, consisting mainly of vegetable oils, are generally rich in unsaturated fatty acids. The latter can be monounsaturated (MUFA) or polyunsaturated (PUFA). Replacing SFA with unsaturated fatty acids (especially PUFA) reduces LDL-C without affecting HDL-C and TG [21]. Additionally, the PBD is characterized by the presence of phytosterols (found in all products of plant origin), which reduce the absorption of cholesterol [22]. Wang et al. conducted a meta-analysis showing that PBD significantly lowered total, LDL, and HDL cholesterol; however, no effect on triglyceride levels was observed [23]. Identical results were obtained in a meta-analysis evaluating 39 studies [24]. However, there are studies suggesting that TG concentrations are significantly lower in vegetarians than in omnivores [25]. Interestingly, dietary fiber is inversely related to TG concentrations, as demonstrated by Hannon et al. in a crossover study of 117 participants who experienced a reduction in TG levels after 7 days of treatment by means of a controlled diet [26]. There is strong evidence suggesting that vegetarian and particularly vegan dietary patterns have a positive impact on fasting and postprandial blood lipid levels, comparable to the effects achieved through conventional therapeutic diets and statin treatment [23].

Diabetes is a disease that is increasing in prevalence, carrying a significant health and economic burden [27]. Therefore, preventive measures aimed at stopping the “diabetes epidemic” are desirable in public healthcare. The incidence of type 2 diabetes (T2DM) appears to be relatively low among those following a PBD [28]. Vegetarian diets have been shown in several clinical studies to be helpful in the prevention and treatment of type 2 diabetes [29]. Scientific studies have demonstrated a decrease in plasma glucose levels, and, thus, a dramatic reduction in the use of antidiabetic drugs in response to the adaptation of a plant-based diet [30]. A 2014 meta-analysis found that a plant-based diet significantly improved blood sugar control. This conclusion was inferred from lowering the plasma glycated hemoglobin in T2DM patients [31]. One of the mechanisms possibly responsible for the improvement in glycemic control is the increased insulin sensitivity that is achieved by the consumption of soybeans (a common protein replacement in a plant-based diet that contains high amounts of lysine, leucine, phenylalanine, phosphate, and calcium) [32]. Cereals reduce the risk of diabetes because they are rich in magnesium, a deficiency of which impairs the signaling of the insulin pathway [33]. The protective antidiabetic effect of PBD also originates from the lack of fats of animal origin and the increased consumption of foods with a low glycemic index [34]. It is important to remember that the reduction in calorie intake associated with meat-free diets can result in weight loss, which is a widely recognized factor that significantly affects the control of blood sugar levels [31].

## 3. Mediterranean Diet

The Mediterranean diet (MD) is well-researched healthy diet pattern and it is known for its high content of plant foods (fruits, vegetables, legumes, nuts) and olive products. This diet includes an average intake of fish and dietary products and the moderate consumption of alcohol, as during the repast a glass of wine is habitually drunk; red meat and sweets are also eaten occasionally. By the type of products consumed, this diet may be an instrument to control cardiovascular risk factors, such as diabetes, hypertension and hypercholesterolemia [35].

Olive oil has a special place in the Mediterranean diet, of which it is the basis. It has known cardiovascular health benefits, including those on blood pressure, cholesterol level, and thrombogenesis [36,37,38]. Dub et al. proved that olive leaf extract (OLE) has an effect on thrombus morphology—in the groups treated with the extract, the thrombus was filamentous and thin, while in the control group of the rabbit model of thrombosis, the blood clot was thick and completely occluded the vein [39]. In the presented study, OLE significantly prolonged PT without affecting APTT, indicating that the antithrombotic activity of OLE may be due to modification of the extrinsic but not intrinsic coagulation system. Oleuropein, which is the main component of OLE, was found to stimulate the production of nitric acid (NO) in mouse macrophages and activate the inducible form of NO synthase [40]. The inhibition of FVII can be explained by the direct action of nitric oxide (NO) downregulating the factor VII gene [41]. The phenols in OLE probably contribute to the reduction in thrombus adhesion to the vascular walls by inhibiting the homocysteine-induced adhesion of endothelial cells, regardless of their antioxidant activity [42]. Other ingredients that have proven anticoagulant effects are Mediterranean spices, in particular rosmarinic acid (RA). This naturally occurring phenol acid works by inhibiting vascular smooth muscle cell (VSMC) proliferation induced by PDGF reducing neointima formation [43].

Plant-based diets and the Mediterranean diet belong to anti-inflammatory diets [44] (Table 1) that have a proven anticoagulant effects [45,46]. Yuan et al., in the analysis of two population-based cohorts, suggested that a diet with a high anti-inflammatory potential partially offsets the cardiovascular risk caused by smoking [47]. They found statistically significant interactions between the anti-inflammatory diet index (AIDI) and smoking.

## 4. Iron

Iron is an element used in the synthesis of oxygen-transport proteins such as hemoglobin and myoglobin, as well as in the formation of heme enzymes and other iron-containing enzymes which are involved in oxidation and reduction [48]. More than 50% of body iron is found in hemoglobin in circulating erythrocytes. A proportion of 25% is found in easily mobilized iron stores, and the remainder is bound to myoglobin in muscle tissue and various enzymes involved in oxidative metabolism [49].

In the eye, iron plays a key role in retinal metabolism and phototransduction. RPE-65 is an iron-containing protein expressed in the retinal pigment epithelium. It is an enzyme (isomerohydrolase) that is required in the catalytic conversion of all-trans-retinyl ester to 11-cis-retinol, a critical step in the visual cycle [50]. RPE-65 is also important for maintaining the excitability of photoreceptors through the phagocytosis of exfoliated outer photoreceptor segments, allowing for the reconstruction of their photosensitive outer layer [51,52]. Moreover, iron is essential for guanylate cyclase for the synthesis of cGMP, the second messenger in the phototransduction pathway. Photoreceptor regeneration also depends on iron-containing enzymes, such as fatty acid desaturase, required for the synthesis of lipids necessary in membranes [53].

The body relies solely on the diet as a source of iron, which exists in two forms: heme and non-heme. Heme iron is derived from hemoglobin and other hemoproteins, primarily found in animal products. On the other hand, iron in plant-based foods is in the form of non-heme iron, which is absorbed less efficiently in the intestines compared to heme iron [54]. The main cause of iron deficiency is primarily the low intake of bioavailable iron, as well as inadequate intestinal absorption, excessive blood loss, an increased need for iron as a result of rapid growth, pregnancy or due to specific genetic mutations [55]. Iron deficiency in the body results in changes in the structure of red blood cells (they become smaller), which leads to microcytic anemia. Iron deficiency anemia (IDA) is the most common form of anemia and a known cause of large vessel occlusion [56]. Retinal venous circulation is characterized by high vascular resistance and low blood flow rate, making it particularly vulnerable to the formation of thrombi as a result of hematological changes [57,58]. The basic mechanism of IDA thrombosis is most likely the result of the direct process of reactive thrombocytosis and other mechanisms consistent with the principles of Virchow’s triad, and mainly endothelial damage and hypercoagulability. Iron regulates the number and function of platelets by inhibiting thrombopoiesis. In the state of iron deficiency, reactive thrombocythemia occurs which leads to hypercoagulability [59]. Red blood cell deformability is reduced in microcytic cells which results in increased blood viscosity [60]. It has been hypothesized that hypoxia caused by anemia may lead to endothelial cell damage in the choroidal retina [61]. There are many examples of central retinal vein thrombosis induced by iron deficiency anemia in the scientific literature [62,63,64,65].

Iron deficiency is the most prevalent nutrient deficiency in the world, and the growing popularity of plant-based diets reinforces this trend [66]. Cross-sectional studies and descriptive reviews have shown that iron levels in vegetarians are compromised by the lack of highly bioavailable heme iron in meat-free diets and the inhibitory effect of certain plant-based ingredients on the bioavailability of non-heme iron; at the same time, female vegans of reproductive age are primarily at risk of iron deficiency anemia [66,67]. Many plant foods are high in iron, most notably legumes, beans, whole grains, and dark green leafy vegetables [68]. In Western countries, vegetarian diets may contain as much or even more iron as diets that include animal meat [69]. The total content of this element in the diet, however, provides little information on the amount of bioavailable iron, and its bioavailability may vary up to 10 times depending on the diet model being adopted (Table 2) [70]. This is largely due to inhibitory factors–primarily phytic acid found in lentils, nuts, legumes, and whole grains, but also polyphenols such as chlorogenic acid and tannic acid found in vegetables, cereals, tea, coffee, and red wine [71]. At the opposite extreme, there are substances that enhance the bioavailability of non-heme iron, which include ascorbic acid, retinol, and carotenes [72,73]. It is likely that iron absorption from a vegetarian diet can be enhanced to some extent by making changes to food choices and preparation methods. These modifications might involve utilizing iron cookware, which can increase the bioavailability of iron by as much as 9% [74]. To enhance iron absorption, it is recommended to consume iron-rich foods alongside sources of ascorbic acid and carotenes. At the same time, it is advisable to avoid iron-inhibiting foods like coffee or tea between meals and opt for foods with lower levels of phytate.

The relationship between the occurrence of RVO and IDA is one of the strongest. Particularly young women following a PBD should remember to control their blood counts.

## 5. Homocysteine, B Vitamins, and Folic Acid

Homocysteine is a non-protein sulfur-containing amino acid, synthesized in the body from methionine, which is considered an exogenous amino acid, meaning it must be obtained through dietary sources. Meat products are the primary source of methionine, while individuals following meat-free diets, such as vegetarians, can turn to legumes like beans, peas, or lentils as an alternative. Although legumes are plant-based, they do contain smaller amounts of methionine compared to animal-based foods. Vitamin B12 is one of the cofactors required for homocysteine metabolism. It is found in foods such as meat, eggs, fish, milk, and cheeses. Vitamin B12 is an essential micronutrient that plays a crucial role in various biochemical processes, including red blood cell formation, nervous system function, and the biosynthesis of neurotransmitters [75]. Elevated serum homocysteine levels may be attributed to various factors, including genetic abnormalities affecting the enzymes involved in homocysteine metabolism like MTHFR677T gene mutation, deficiencies in essential vitamin cofactors, underlying conditions like chronic renal failure, certain medications such as nicotinic acid and fibrates, pregnancy, and smoking. Elevated homocysteine levels have been associated with prothrombotic and atherogenic properties. [76]. Homocysteine-associated vascular lesions have been observed to include intimal thickening, rupture of the elastic lamina, smooth muscle hyperplasia, and thrombotic occlusion formation [77,78]. Hyperhomocysteinemia has been identified as a possible risk factor in the etiopathogenesis of diseases such as coronary artery disease and stroke [79].

Of the numerous studies that have been conducted on the effects of homocysteine concentration on the retinal venous system, some have shown a link between hyperhomocysteinemia and retinal vein occlusion [80,81,82,83,84,85,86,87,88,89], while other studies have failed to show that hyperhomocysteinemia is an independent risk factor for retinal vein occlusion [90,91,92,93,94]. A meta-analysis conducted by Cahill et al. on a group of 614 patients diagnosed with retinal vein occlusion provided conclusive evidence that there is a clear association between retinal vascular occlusion and increased levels of homocysteine in the plasma, as well as reduced levels of folate in the serum. However, the analysis did not establish a significant correlation between serum vitamin B12 levels and retinal vein occlusion [95]. An interesting study was presented by Francesco Sofia et al., who demonstrated on 262 patients with RVO that low levels of vitamin B6 and folic acid, and elevated levels of homocysteine are independently associated with an increased risk of retinal vein occlusion, while the level of vitamin B12 is not statistically significant [96]. Despite the findings of Cahill and Sofi’s research, other studies have indicated a connection between low levels of vitamin B12 and the occurrence of retinal vein occlusion [97,98].

There are specific interactions between Hcys, B vitamins, and folates. Homocysteine can be metabolized into cystathionine through the transsulfuration pathway, which relies on the presence of vitamin B6. Alternatively, it can be converted back to methionine with the help of vitamin B12 and folic acid through the process of transmethylation [95]. What is more, vitamin B6 may determine the increased risk of thrombotic events not only by catalyzing enzymatic reactions related to homocysteine but also by affecting platelet aggregation [99,100]. Numerous studies show a negative correlation between the level of vitamin B12, B6, and folic acid in the serum and the level of homocysteine in the serum [82,90,101,102].

Hyperhomocysteinemia may, therefore, be a potentially modifiable risk factor for retinal vein occlusive and a simple intervention such as vitamin B12, B6, and folic acid supplementation could possibly address this risk factor [101,103,104]. Nevertheless, folate levels can be increased through the consumption of leafy vegetables and wholegrain cereals. Obtaining folate from natural food sources may offer benefits compared to synthetic folic acid found in supplements, as the latter has been linked to a potential increased risk of prostate cancer [105]. Dietary folate occurs in a reduced form with polyglutamate chains requiring oxidation and hydrolysis for absorption. On the other hand, folic acid is already present in its oxidized form, known as oxidized pteroylmonoglutamate, which grants it approximately 50% higher bioavailability [105]. Research has shown that approximately 5% of the US population surpasses the recommended upper limit of folate intake, primarily due to the usage of dietary supplements. Interestingly, among adults with a history of prostate cancer, the use of dietary supplements and multivitamin-multimineral (MVMM) products is significantly higher compared to the general population (56% compared to 35–40% for the healthy population for MVMM) [106]. Moreover, the National Cancer Institute of the USA has identified folic acid as a potential risk factor for prostate cancer when consumed in large amounts as a supplement [107].

Consuming meals rich in B vitamins and folic acid can lower plasma homocysteine levels by 15% and raise the levels of serum folic acid [108,109]. Towards the end of the 20th century, the US and Canadian governments mandated the fortification of cereal products with folic acid to reduce the risk of neural tube defects [110,111]. This process led to more than a 90% decrease in the incidence of folic acid deficiency and a 50% decrease in the incidence of elevated homocysteine levels in the study population [112]. Dhipak A. et al. point out that the high incidence of vitamin B12 deficiency reported in the Indian population may be due to general dietary habits, which include factors such as vegetarianism, overcooking, and the aforementioned lack of fortification of cereals [90]. Smoking and excessive consumption of alcohol and caffeine are other modifiable dietary risk factors for elevated plasma homocysteine levels [113,114,115].

Numerous studies suggest that maintaining a balanced homeostasis between homocysteine and its cofactors is crucial for the optimal health of retinal blood vessels. However, it is important to bear in mind that excessive and unregulated supplementation of folic acid and vitamin B12 has been proven to have carcinogenic effects.

## 6. Vitamin D, A, C, and Potassium

There are multiple sources of vitamin D available to us: local production in the skin, dietary intake, and supplementation. Two essential forms of vitamin D are cholecalciferol (vitamin D3) and ergocalciferol (vitamin D2). The richest natural sources of cholecalciferol include the flesh of fatty fish and fish oils, such as sardines, mackerel, eel, cod, salmon, and tuna [116]. On the other hand, dried mushrooms are a notable source of vitamin D2 (Table 3) [116]. In order to prevent possible vitamin D deficiencies, some food products are fortified with it. In the USA and in European countries such products are mainly milk, margarine, yogurt, breakfast cereals, and others [116]. Endogenous production in the skin is stimulated by exposure to UV radiation with a length of 290–315 nm, which in the final stage enables the conversion of biologically inactive vitamin D2 and D3 into the biologically active form of vitamin D, namely 1α,25-dihydroxyvitamin D (1,25(OH)2D) (calcitriol) [117]. The chemical processes leading to the creation of the active form of the vitamin, i.e., calcitriol, take place in the skin, liver, and kidneys.

The concept of vitamin D being solely classified as a vitamin should be discarded. Vitamin D is a versatile hormone that serves multiple functions and is recognized for its crucial involvement in various metabolic processes beyond its traditional role in calcium regulation. Vitamin D receptors are found in a wide range of tissues that extend beyond those involved in calcium metabolism. These include hepatocytes, lymphocytes, cardiac cells, and vascular myocytes [118]. Recent research suggests that enzymes responsible for hydroxylating vitamin D are also found in various eye structures, including the cornea, ciliary body, sclera, and retina. This observation indicates that vitamin D might play a crucial role as an intraocular mediator in eye-related disorders [119,120]. In addition, more and more studies indicate that vitamin D, its receptors (VDR), and calcitriol analogs may play an important role in the proper functioning of the eye [121,122].

Vitamin D deficiency is associated with an increased risk of ischemic stroke, coronary heart disease, and venous thromboembolism, and is associated with high mortality from vascular events involving the coronary and cerebral circulation [123,124,125]. Interestingly, several studies have shown a seasonal variability in the incidence of cardiovascular disease, with a higher incidence in the winter [126,127]. Several studies have suggested a seasonal onset of RVO and a higher incidence of retinal vein occlusion in winter [128,129]. Serum vitamin D concentrations are known to fluctuate greatly with the seasons with a peak occurring in summer [130]. Numerous studies on vitamin D levels in retinal vein occlusion have been published. One of the first scientists to draw attention to the possible association of vitamin D with retinal vein occlusion was Talcott et al. [131]. Epstein analyzed 72 patients with central retinal vein occlusion and found no statistically significant differences in the level of vitamin D compared to the control group, but demonstrated that 51.4% of patients with CRVO had a deficiency of this vitamin in comparison to 39.3% of patients in the control group [132]. Another study in Indian patients indicates a potential role of vitamin D in retinal vein occlusion [133]. An interesting result was presented by Saeed Karimi et al., who proved that oral vitamin D supplementation can improve the anatomical and functional parameters of the macula in patients with macular edema secondary to central retinal vein occlusion (CRVO) treated with intravitreal bevacizumab [134].

It is worth reiterating that systemic vascular diseases and retinal vascular occlusion share several risk factors. In vitro studies have shown that the interaction of vitamin D with the VDR increases the formation of nitric oxide, thereby reducing oxidative stress [135]. Another non-classical function of vitamin D is to suppress the activation of cytokines and reduce the proliferation of inflammatory cells by modulating the immune system. It is confirmed that vitamin D has the potential for an anti-inflammatory response, mainly by inhibiting the nuclear factor kapa B (NF-κB) signaling pathway and inhibiting pro-inflammatory factors [122]. In addition, it was found that vitamin D modulates apoptosis and reduces the expression of VEGF, inhibiting angiogenesis [136]. Vitamin D counteracts oxidative stress and reduces inflammation by reducing the expression of the protein interleukin 1, interleukin 8 and TNF-α. The listed anti-inflammatory, antioxidant and anti-angiogenic effects of vitamin D confirm its possible effect on the occurrence of RVO in conditions of its deficiency. These mechanisms contribute to stimulating the anti-atherosclerotic immune response [137,138]. Finally, low vitamin D levels are associated with increased vascular resistance and hypertension that is secondary to a lack of inhibition of the renin-angiotensin system responsible for blood pressure regulation [139]. The lack of vitamin D in the context of systemic vascular risk, endothelial homeostasis, and inflammation is likely implicated in the pathogenesis of retinal vein occlusion (RVO).

Supplementing vitamin D deficiency is technically simple (diet, dietary supplements, sun exposure for 30 min between 10 a.m. and 2 p.m. in accordance with WHO recommendations), which may facilitate further prevention of cardiovascular diseases [140].

Vitamins A, C, and potassium are primarily obtained from fruits and vegetables. Incorporating fruits and vegetables into one’s diet, as they are rich in these nutrients, along with polyphenols and carotenoids, plays a significant role in the prevention and management of eye diseases. These compounds possess various beneficial properties, such as anti-inflammatory, antioxidant, and anti-angiogenic effects, which contribute to overall health and well-being [141].

The study conducted by Gopintah et al. confirmed the hypothesis that a nutrient-rich diet, particularly one abundant in vegetables and fruits, is linked to a narrower retinal venular caliber. This narrower caliber suggests a higher quality of the retinal microcirculation [142]. Ayaka Edo showed that an increased intake of vitamins A, C, and potassium results in a decrease in the caliber of retinal veins, but no effects on arterial vessels have been demonstrated [143]. Deviations from the optimal retinal vascular structure are thought to include narrower retinal artery size and wider retinal vein size, which are independent predictors of ischemic heart disease and stroke [144,145]. Past studies have shown that supplementation with antioxidant vitamins A, C, and adequate potassium in the diet contributes to reducing the risk of CHD and stroke [146,147]. Arterial stenosis and venous enlargement are both considered indicators of systemic microcirculation deterioration. However, recent research has highlighted the independent role of veins, particularly a wider venular caliber, as a potentially significant marker of microvascular disease [148]. Notably, smoking has been specifically linked to a wider venular caliber [149]. Tamai et al. conducted a study involving the injection of lipid hydroperoxide into the vitreous body of rats, which resulted in increased leukocyte count in the retinal microcirculation and an enlargement of the retinal venular caliber, suggesting a connection between oxidative stress and these vascular changes [150]. Epidemiological studies also show that systemic markers of inflammation such as white blood cell count, c-reactive protein (CRP), serum amyloid A, and interleukin-6 are associated with a large retinal venular caliber [151]. Experimental studies have also shown that vitamins A and C inhibit the activation of inflammatory promoters [152]. Vitamin A and C have proven abilities to capture reactive oxygen species and inhibit the activation of NF-κB, a transcription factor that promotes the expression of inflammation-inducing genes [153]. While vitamins A and C are commonly recognized as antioxidant nutrients, it is worth noting that dietary potassium also possesses antioxidant properties and can mitigate the formation of free radicals [154].

While there is no direct evidence establishing the presented nutrients as inhibitors of retinal vein occlusion, the studies presented provide strong indications that these nutrients have a beneficial impact on the “quality” of retinal venous vessels. This suggests that they may offer protection against retinal occlusive events.

## 7. Coenzyme Q10

Coenzyme Q10, a vitamin-like compound, undergoes a decline in body concentration with age. It functions within the inner mitochondrial membrane, playing a crucial role in generating adenosine triphosphate (ATP) via the electron transport chain. Additionally, it acts as an antioxidant, safeguarding cells against oxidative stress caused by reactive oxygen species (ROS). Furthermore, it helps maintain a proton (H+) gradient across lysosomal membranes, aiding in the breakdown of cellular waste materials [155]. At the cellular level, CoQ10 deficiency may result in apoptosis, which can be visualized in tissues of the central nervous system (CNS). The positive effect of using coenzyme Q10 in neurodegenerative diseases of the brain (Huntington’s disease and Friedrich’s Ataxia) and its deficiency in age-related macular degeneration has been demonstrated [156].

Fernandez-Vega et al. presented a compelling study that investigated the use of vitamins and coenzyme Q10 for treating retinal dysfunction attributed to vascular disorders. Their research showcased the therapeutic potential of coenzyme Q10, as it resulted in a substantial improvement in the visual field, highlighting its effectiveness in enhancing vision [157]. Unfortunately, none of the 48 patients included in the study had retinal vein occlusion (RVO). However, the positive therapeutic outcome observed in patients with central retinal artery occlusion provides optimism that the utilization of CoQ10, in conjunction with conventional treatment approaches, could enhance the treatment outcomes for individuals with RVO. To validate these findings, randomized clinical trials are necessary.

## 8. COVID-19 Pandemic and Dietary Habits

The outbreak of the SARS-CoV-2 virus and the ensuing pandemic have compelled individuals to alter their dietary habits. As a result of the COVID-19 pandemic, numerous countries have implemented measures such as the closure of food establishments and the implementation of stay-at-home orders [158]. Psychological distress has driven individuals to develop unhealthy behaviors, including reduced physical activity and an unhealthy diet. This is evident from the rise in alcohol consumption and the consumption of sugary foods. Consequently, there has been a shift in the balance between calorie intake and energy expenditure [159]. In addition, numerous studies have indicated a rise in the consumption of easily prepared food items, including animal-based products and canned foods, during the COVID-19 pandemic [160]. Further investigations have demonstrated substantial decreases in the intake of fruits and vegetables, primarily due to restricted access and inflated prices associated with these food items. As a consequence, adherence to the Mediterranean diet has been adversely affected, resulting in poor compliance with its recommended dietary patterns [161]. Moretti et al. in their review showed that the change in the quality and quantity of macro and micronutrient intake is related to increased bone fragility during the COVID-19 pandemic [162].

Numerous publications have provided evidence of the distinct impact of SARS-CoV-2 infection on the elevation of blood clotting [163,164]. To access target cells, the SARS-CoV-2 virus utilizes the angiotensin-2 receptor (ACE). Interestingly, these specific receptors have been identified in various ocular structures, including Muller cells of the retina, retinal blood vessels, choroid, conjunctiva, and corneal epithelial cells [165,166]. Fonollosa et al. conducted a noteworthy study that examined 15 patients with COVID-19 accompanied by retinal vascular occlusion. Their collective analysis revealed that central retinal vein occlusion was the most common type of occlusion observed in this group. Additionally, the study indicated that patients with COVID-19-related retinal vascular occlusion tended to be younger (with a median age of 39 years) compared to non-COVID patients [167].

The demonstrated impact of the SARS-CoV-2 virus on dietary habits, lifestyle, and increased coagulability clearly suggests a potential link to retinal vein occlusion (RVO).

## 9. Summary

Very few direct studies investigating the impact of nutrition on the occurrence of retinal vein occlusion exist. Given the prevalence of this disease and its significant impact on visual impairment, further research is necessary to comprehensively understand the effects of nutrients on the retinal microcirculation system. Numerous studies have demonstrated the potential benefits of nutrients, emphasizing the importance of a well-balanced diet as a valuable nutraceutical. Plant-based diets and the Mediterranean diet reduce the most important RVO risk factors: hypertension, hyperlipidemia and diabetes. These dietary patterns belong to anti-inflammatory diets that have a proven anticoagulant effect. A diet containing a low amount of heme iron in predisposed people changes the structure of red blood cells, leading to hypercoagulability. The deficiency of B vitamins and folic acid may be independent risk factors for RVO. For this reason, especially in burdened people, we should take into account the level of these microelements in the blood. The listed vitamins with antioxidant and anti-inflammatory properties such as vit. A, C or Vit. D have a proven inhibitory effect on cytokines and chemokines involved in the pathogenesis of RVO. It is essential to remember that each diet should be carefully planned and monitored by a dietitian to prevent potential nutritional deficiencies and avoid the excessive intake of macro and micronutrients, which can have adverse effects. The COVID-19 pandemic has disrupted lifestyle habits, including dietary patterns, which has been acknowledged as a challenge by the WHO. Consequently, the organization has issued food and nutrition guidelines to assist individuals or families in meeting their nutritional needs during quarantine periods [168].

## 10. Method of Literature Search

Pubmed was searched with no year limitations. The following keywords were used: “diet”, “vitamin” and “nutrients”. Each of these keywords was used in combination with the expression “retinal vein occlusion”. Further searches were conducted combining the stated keywords with epidemiology, mechanisms, pathogenesis, neovascularization, cytokines, chemokines, plant-based diets, Mediterranean diet and treatment. After reviewing the available literature, we included relevant information on dietary habits and its impact on retinal vein occlusion.

## Figures and Tables

**Figure 1 nutrients-15-03237-f001:**
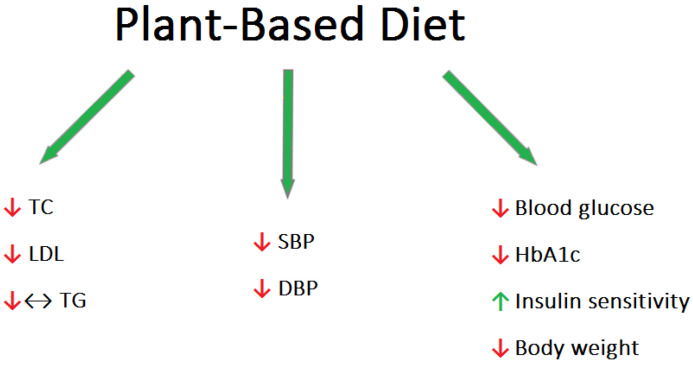
Main protection mechanisms of the PBD against RVO [15].

**Table 1 nutrients-15-03237-t001:** Potential anti-inflammatory protective mechanisms of a plant-based diet (PBD) and Mediterranean Diet (MD) against RVO [44].

PBD	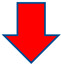	CRP, TNF-α, TNFR-60, IL-1, IL-4, IL-6, fibrinogen, sE-selectin
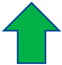	γδ-T cell populations
MD	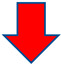	CRP, TNF-α, TNFR-60, IL-1, IL-6, IL-7, IL-8, IL-10, IL-13, IL-18, VEGF, MMP-9, MCP-1, sVCAM-1, sICAM-1, PAI-1, leptin
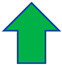	Adiponectine

CRP: C-reactive protein; IL: Interleukin; MCP-1: Monocyte chemoattractant protein 1; MMP-9: Matrix metallopeptidase-9; PAI-1: Plasminogen activator inhibitor 1; sICAM-1: Soluble intercellular adhesion molecule 1; sVCAM-1: Soluble vascular cell adhesion molecule 1; TNF-α: Tumor necrosis factor; TNFR: Tumor necrosis factor receptor; VEGF: Vascular endothelial growth factor.

**Table 2 nutrients-15-03237-t002:** Factors that could influence iron absorption, potentially increasing or decreasing the risk of RVO [70].

Enhancers	Inhibitors
Meat, poultry, fish	Phytic acid
Ascorbic Acid	Polyphenols
Retinol and carotenes	Calcium and proteins in milk products
Alcohol	Egg
	Soy protein

**Table 3 nutrients-15-03237-t003:** Vitamin D content of various foods [116].

Vitamin D Content in Food Products	
Name of the Product	Vitamin D (µg/100 g)
Mackerel chilled/frozen, raw, flesh only	8.0
Salmon, raw	5.0
Sardines chilled/frozen, raw, flesh only	4.0
Yellowfin tuna chilled/frozen, raw, flesh only	3.2
Cod chilled/frozen, raw, flesh only	Trace
Prawns, king, raw	Trace
Eggs, chicken, whole raw	3.2
Pork leg joint, raw	0.9
Lamb chop, raw	0.8

## Data Availability

Not applicable.

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
