# Peer review of "Prevention and Treatment of Retinal Vein Occlusion: The Role of Diet—A Review"

_nutrients, 2023, doi:10.3390/nu15143237_

Round 1
Reviewer 1 Report
The author attempts to review the role of diet and lifestyle in preventing retinal vein occlusion (RVO) in this article. I have the following suggestions for the authors of this article:
The article discusses the impact of dietary habits on RVO, mainly exploring the impact of plant-based diets on retinal related risk factors. It is not comprehensive enough, and it is necessary to comprehensively elaborate on the impact of various dietary habits on RVO in order to have broad reference value.
The discussion of lifestyle in the article is only limited to the changes in lifestyle during the COVID-19 pandemic, which is too narrow.
The focus of the article is actually the impact of various nutritional elements on RVO, so whether the title of the article needs to be modified to more accurately express the content of the article needs further consideration by the author.
The article can provide a more comprehensive overview of relevant experimental research, explore related molecular mechanisms, and make the article more convincing.
Minor editing of English language required
Author Response
Thank you very much for reviewing our manuscript. We wish to express our appreciation to you for your useful comments, which have helped us to improve our manuscript. According to the suggestions, we have revised our manuscript. We hope that you find our responses satisfactory.
Response to reviewer 1:
The author attempts to review the role of diet and lifestyle in preventing retinal vein occlusion (RVO) in this article. I have the following suggestions for the authors of this article:
Reviewer wrote:
1.The article discusses the impact of dietary habits on RVO, mainly exploring the impact of plant-based diets on retinal related risk factors. It is not comprehensive enough, and it is necessary to comprehensively elaborate on the impact of various dietary habits on RVO in order to have broad reference value.
Response: Thank you for your remark. We added some important informations.
Mediterranean diet ( MD) is well researched healthy diet pattern and it is known for its high content of plant foods ( fruits, vegetables, legumes, nuts) and olive products. This diet includes average intake of fish, dietary products and moderate consumption of alcohol but during the repast a glass of wine is habitually drunk, red meat and sweets are eaten occasionally. By the type of products consumed, this diet may be an instrument to control cardiovascular risk factors, such as diabetes, hypertension and hypercholesterolemia [1].
Olive oil has a special place in the Mediterranean diet, of which it is the basis. It has known cardiovascular health benefits, including those on blood pressure, cholesterol level, and thrombogenesis [2,3,4]. Dub et.al proved that olive leaf extract (OLE) has an effect on thrombus morphology - in the groups treated with the extract, the thrombus was filamentous and thin, while in the control group of the rabbit model of thrombosis, the blood clot was thick and completely occluded the vein [5]. In the presented study, OLE significantly prolonged PT without affecting APTT, indicating that the antithrombotic activity of OLE may be due to modification of the extrinsic but not intrinsic coagulation system. Oleuropein, which is the main component of OLE, was found to stimulate the production of nitric acid (NO) in mouse macrophages and activate the inducible form of NO synthase [6]. The inhibition of FVII can be explained by the direct action of nitric oxide (NO) downregulating the factor VII gene [7]. The phenols in OLE probably contribute to the reduction of thrombus adhesion to vascular walls by inhibiting homocysteine-induced adhesion of endothelial cells, regardless of their antioxidant activity [8]. Other ingredients that have proven anticoagulant effects are mediterranean spices, in particular rosmarinic acid (RA). This naturally-occurring phenol acid works by inhibiting vascular smooth muscle cel (VSMC) proliferation induced by PDGF reducing neointima formation [9].
Plant-based diets and Mediterranean diet belong to anti-inflammatory diets that have a proven anticoagulant effect [10,11]. Yuan et al. in the analysis of two population-based cohorts suggested, that a diet with a high anti-inflammatory potential partially offsets the cardiovascular risk caused by smoking [12]. They found statistically significant interaction between anti-inflammatory diet index (AIDI ) and smoking.
- Fitó, M.; Konstantinidou, V. Nutritional Genomics and the Mediterranean Diet’s Effects on Human Cardiovascular Health.Nutrients 2016, 8, 218, doi:10.3390/nu8040218.
- Khayyal, M.T.; El-Ghazaly, M.A.; Abdallah, D.M.; Nassar, N.N.; Okpanyi, S.N.; Kreuter, M.H. Blood Pressure Lowering Effect of an Olive Leaf Extract (Olea Europae) in L-NAME Induced Hypertension in Rats.Arzneimittelforschung 2002, 52, 797–802, doi:10.1055/s-0031-1299970.
- Fki, I.; Bouaziz, M.; Sahnoun, Z.; Sayadi, S. Hypocholesterolemic Effects of Phenolic-Rich Extracts of Chemlali Olive Cultivar in Rats Fed a Cholesterol-Rich Diet. Med. Chem. 2005, 13, 5362–5370, doi:10.1016/j.bmc.2005.05.036
- Brzosko, S.; De Curtis, A.; Murzilli, S.; de Gaetano, G.; Donati, M. B.; Iacoviello, L. Effect of Extra Virgin Olive Oil on Experimental Thrombosis and Primary Hemostasis in Rats. Metab. Cardiovasc. Dis. 2002,12, 337–342.
- Dub, A.M.; Dugani, A.M. Antithrombotic Effect of Repeated Doses of the Ethanolic Extract of Local Olive (Olea Europaea L.) Leaves in Rabbits.Libyan J. Med. 2013, 8, 20947, doi:10.3402/ljm.v8i0.20947.
- Visioli, F.; Bellosta, S.; Galli, C. Oleuropein, the Bitter Principle of Olives, Enhances Nitric Oxide Production by Mouse Macrophages.Life Sci. 1998, 62, 541–546, doi:10.1016/s0024-3205(97)01150-8.
- De Lucas, S.; Bartolome, J.; Carreno, V. Nitric Oxide Downregulates Factor VII Gene by Inhibiting the Binding of SP1 and HNF-4. Hepatol. 2003, 38, 75–76.
- Manna, C.; Napoli, D.; Cacciapuoti, G.; Porcelli, M.; Zappia, V. Olive Oil Phenolic Compounds Inhibit Homocysteine-Induced Endothelial Cell Adhesion regardless of Their Different Antioxidant Activity. Agric. Food Chem. 2009, 57, 3478–3482, doi:10.1021/jf8037659.
- Liu, R.; Heiss, E.H.; Waltenberger, B.; Blažević, T.; Schachner, D.; Jiang, B.; Krystof, V.; Liu, W.; Schwaiger, S.; Peña-Rodríguez, L.M.; Breuss, J.M.; Stuppner, H.; Dirsch, V.M.; Atanasov, A.G. Constituents of Mediterranean Spices Counteracting Vascular Smooth Muscle Cell Proliferation: Identification and Characterization of Rosmarinic Acid Methyl Ester as a Novel Inhibitor. Nutr. Food Res. 2018, 62, e1700860, doi:10.1002/mnfr.201700860.
- Violi, F.; Pastori, D.; Pignatelli, P.; Carnevale, R. Nutrition, Thrombosis, and Cardiovascular Disease. Res.2020, 126, 1415–1442, doi:10.1161/CIRCRESAHA.120.315892.
- Steffen, L.M.; Folsom, A.R.; Cushman, M.; Jacobs, D.R.; Rosamond, W.D. Greater Fish, Fruit, and Vegetable Intakes Are Related to Lower Incidence of Venous Thromboembolism: the Longitudinal Investigation of Thromboembolism Etiology. Circulation 2007,115, 188–195, doi:10.1161/CIRCULATIONAHA.106.641688
- Yuan, S.; Bruzelius, M.; Damrauer, S.M.; Håkansson, N.; Wolk, A.; Åkesson, A.; Larsson, S.C. Anti-Inflammatory Diet and Venous Thromboembolism: Two Prospective Cohort Studies. Metab. Cardiovasc. Dis.2021, 31, 2831–2838, doi:10.1016/j.numecd.2021.06.021.
Reviewer wrote:
- The discussion of lifestyle in the article is only limited to the changes in lifestyle during the COVID-19 pandemic, which is too narrow.
Response: Thank you for your suggestion. Indeed, the title could mislead the reader. Corrected the title. Now It is "Prevention and Treatment of Retinal Vein Occlusion: The Role of Diet – A Review".
Reviewer wrote:
3.The focus of the article is actually the impact of various nutritional elements on RVO, so whether the title of the article needs to be modified to more accurately express the content of the article needs further consideration by the author.
Response: Thank you for your remark. We changed the title.
Reviewer wrote:
4.The article can provide a more comprehensive overview of relevant experimental research, explore related molecular mechanisms, and make the article more convincing.
Response: Thank you for drawing attention to such an important issue. I will try to briefly describe the molecular relationships occurring in RVO. Some of these micronutrients have proven effects on these signaling pathways.
There is ample evidence that cytokines and chemokines contained in the vitreous body are correlated with the occurrence of RVO, especially the interleukin family, VEGF, MMP, LPA-ATX and PDGF [4]. The interleukin family is pro-inflammatory, causing ischemia and macular edema secondary to RVO, and the most important in this disease include: IL-6, IL-8, IL-17 and IL-18, which trigger STAT3, MAPK, NF-κB, VEGF pathways and provokes ROS. VEGF inhibits occludin by damaging the basement membrane of endothelial cells and activates MMP-9 to destroy blood-retinal barriers (BRB) and induces ICAM-1 causing leukocyte stasis. VEGF also works by activating the NOX1 and NOX4 proteins, which are dominant in ROS production in RVO. MMP-2 and MMP-7 are involved in the migration of vascular endothelial cells. The LPA-ATX signaling pathway may mediate inflammation in RVO as it activates IL-6, IL-8, VEGF and MMP-9. PDGF-A potentiates VEGF to induce neovascularization.
An interesting study was presented by Takuma Neo et al., who used the rabbit retinal vein occlusion model in order to analyze ischemia-induced changes in gene expression profiles [3] . The study revealed that angiogenic regulators Dcn and Mmp1 and the pro-inflammatory factors Mmp12 and Cxcl13 were significantly elevated in RVO retinas. In total, they analyzed 387 genes with more than a 2-fold difference between RVO and controls (upregulated: 333 genes, downregulated: 54 genes).What is more, they confirmed that JAK-STAT, TNFα and NF-κB pathways likely contribute to rabbit RVO pathology and potentially human retinal ischemic disease. - this paragraph will be added in the introduction
It has been confirmed that vitamin D has the potential for an anti-inflammatory response mainly by inhibiting the nuclear factor kapa B (NF-κB) signaling pathway and inhibiting pro-inflammatory factors [1]. In addition, it was found that vitamin D modulates apoptosis and reduces the expression of VEGF, inhibiting angiogenesis [2] . Vitamin D counteracts oxidative stress and reduces inflammation by reducing the expression of the protein interleukin 1, interleukin 8 and TNF-α. The listed anti-inflammatory, antioxidant and anti-angiogenic effects of vitamin D confirm its possible effect on the occurrence of RVO in conditions of its deficiency. - this paragraph will be added in the vitamin D section
- Caban, M.; Lewandowska, U. Vitamin D, the Vitamin D Receptor, Calcitriol Analogues and Their Link with Ocular Diseases.Nutrients 2022, 14, 2353, doi:10.3390/nu14112353.
- El-Sharkawy, A.; Malki, A. Vitamin D Signaling in Inflammation and Cancer: Molecular Mechanisms and Therapeutic Implications.Molecules 2020, 25, 3219, doi:10.3390/molecules25143219.
- Neo, T.; Gozawa, M.; Takamura, Y.; Inatani, M.; Oki, M. Gene Expression Profile Analysis of the Rabbit Retinal Vein Occlusion Model. PLOS ONE2020, 15, e0236928, doi:10.1371/journal.pone.0236928
Vitamin A and C have proven abilities to capture reactive oxygen species and inhibit the activation of NF-κB, a transcription factor that promotes the expression of inflammation-inducing genes [1]. - this paragraph will be added in the vitamin A, C section
- Conner, E. M.; Grisham, M. B. Inflammation, Free Radicals, and Antioxidants.Nutrition 1996, 12, 274–277, doi:10.1016/s0899-9007(96)00000-8.

Reviewer 2 Report
- Line 251 - could the authors elaborate on why folic acid supplements increase the
risk of prostate cancer (compared to natural dietary sources)?
- Line 324 - please cite those WHO recommendations
- I strongly suggest an extended summary section, which should link more closely
the matters discussed with the retinal venous obstructive pathology

Minor editing of English language required
Author Response
Thank you very much for reviewing our manuscript. We wish to express our appreciation to you for your useful comments, which have helped us to improve our manuscript. According to the suggestions, we have revised our manuscript. We hope that you find our responses satisfactory.
Response to reviewer 2:
Reviewer wrote:
- Line 251 - could the authors elaborate on why folic acid supplements increase the
risk of prostate cancer (compared to natural dietary sources)?
Response : Thank you for your suggestion. We added some information explaining the increased risk of prostate cancer when using folic acid supplements compared to natural dietary sources.
Dietary folate occurs in a reduced form with polyglutamate chains requiring oxidation and hydrolysis for absorption. On the other hand, folic acid is already present in its oxidized form, known as oxidized pteroylmonoglutamate, which grants it approximately 50% higher bioavailability [1]. Research has shown that approximately 5% of the US population surpasses the recommended upper limit of folate intake, primarily due to the usage of dietary supplements. Interestingly, among adults with a history of prostate cancer, the use of dietary supplements and multivitamin-multimineral (MVMM) products is significantly higher compared to the general population (56% compared to 35-40% for the healthy population for MVMM)[2]. Moreover, the National Cancer Institute of the USA has identified folic acid as a potential risk factor for prostate cancer when consumed in large amounts as a suplement[3].
- Pieroth, R.; Paver, S.; Day, S.; Lammersfeld, C. Folate and Its Impact on Cancer Risk. Nutr. Rep. 2018, 7, 70–84, doi:10.1007/s13668-018-0237-y.
- Rock, C. L. Multivitamin-Multimineral Supplements: Who Uses Them? J. Clin. Nutr. 2007,85, 277- 279, doi:10.1093/ajcn/85.1.277S.
- Boyles, A.L.; Yetley, E.A.; Thayer, K.A.; Coates, P.M. Safe Use of High Intakes of Folic Acid: Research Challenges and Paths Forward. Rev. 2016, 74, 469–474, doi:10.1093/nutrit/nuw01.
Reviewer wrote:
- Line 324 - please cite those WHO recommendations
Response :
Thank you for your attention. We cited those WHO recommendations.
[122] Expert Consultation on vitamin and mineral requirement in human nutrition: Second Edition FAO Rome. Available at: https://apps.who.int/iris/bitstream/handle/10665/42716/9241546123.pdf;jsessionid=3A893750E1EEE89B2C501C7FAAB728F7?sequence=1
Reviewer wrote:
I strongly suggest an extended summary section, which should link more closely
the matters discussed with the retinal venous obstructive pathology
Response : Thank you for your attention. At the suggestion of the Reviewer, we extended summary section.
Very few direct studies investigating the impact of nutrition on the occurrence of retinal vein occlusion exist. Given the prevalence of this disease and its significant impact on visual impairment, further research is necessary to comprehensively understand the effects of nutrients on the retinal microcirculation system. Numerous studies have demonstrated the potential benefits of nutrients, emphasizing the importance of a well-balanced diet as a valuable nutraceutical. Plant-based diets and Mediterranean diet reduce the most important RVO risk factors: hypertension, hyperlipidemia and diabetes. These dietary patterns belong to anti-inflammatory diets that have a proven anticoagulant effect. A diet containing a low amount of heme iron in predisposed people changes the structure of red blood cells, leading to hypercoagulability. Deficiency of B vitamins and folic acid may be independent risk factors for RVO. For this reason, especially in burdened people, we should take into account the level of these microelements in the blood. The listed vitamins with antioxidant and anti-inflammatory properties such as vit. A, C or Vit. D have a proven inhibitory effect on cytokines and chemokines involved in the pathogenesis of RVO. It is essential to remember that each diet should be carefully planned and monitored by a dietitian to prevent potential nutritional deficiencies and avoid excessive intake of macro and micronutrients, which can have adverse effects. The COVID-19 pandemic has disrupted lifestyle habits, including dietary patterns, which has been acknowledged as a challenge by the WHO. Consequently, the organization has issued food and nutrition guidelines to assist individuals or families in meeting their nutritional needs during quarantine periods [148].
Reviewer 3 Report
The authors present a comprehensive review in their manuscript titled "Prevention and Treatment of Retinal Vein Occlusion: The Role of Diet and Lifestyle". They explore the potential role of a plant-based diet and lifestyle modifications in preventing and treating retinal vein occlusion (RVO). Moreover, the authors discuss the efficacy of micronutrients such as iron, homocysteine, vitamins, folic acid, potassium, and coenzyme Q10 in improving vision. This review aims to provide valuable insights to the readers regarding nutritional and lifestyle strategies for RVO. However, there are certain areas that should be further enhanced to augment the manuscript's impact and clarity.
1. Indeed, the title and abstract may create the impression that the manuscript is a research article rather than a review. To better reflect the nature of the paper, it would be advisable to modify the title and abstract to clearly indicate that it is a review article. For example: Title should be "A Comprehensive Review on the Prevention and Treatment of Retinal Vein Occlusion: The Role of Diet and Lifestyle" or "Prevention and Treatment of Retinal Vein Occlusion: The Role of Diet and Lifestyle – A Review".
2. The last sentence of the abstract, "The aim of this study is to check whether macro and micronutrients can affect retinal vein occlusion," does not seem suitable for a review paper. It may imply that the current manuscript is a research article. It would be better to rephrase it as follows: "The aim of this review is to investigate the potential impact of macro and micronutrients on retinal vein occlusion".
3. The manuscript predominantly focuses on dietary habits for the prevention and treatment of RVO, with limited/no information provided about lifestyle changes. However, the title of the manuscript mentions both diet and lifestyle.
4. It would be beneficial to the audience if authors briefly discuss the range of treatment strategies currently available for RVO, beyond dietary and lifestyle adjustments, like anti-VEGF therapy, corticosteroids, laser therapy, intravitreal implants, retinal surgery, and more.
5. Several genes have been implicated in the pathogenesis of RVO. It would be informative for the authors to explore whether the expression of these genes is influenced by the macro and micronutrients discussed in the manuscript. To enhance comprehension of the effect of nutrients on RVO pathogenesis, the authors may consider presenting a table or schematic representation that lists the genes known to be affected.
6. It would be valuable to provide the rationale for selecting the specific list of micronutrients discussed in this manuscript. Including this information at the appropriate place within the manuscript would enhance the understanding of why these particular micronutrients were chosen for discussion.
7. The content in the subheading "Covid-19 pandemic and lifestyle" appears to be unrelated to the context of the manuscript. It would be advisable to either remove or revise this subheading to ensure its alignment with the overall focus and scope of the manuscript.
8. Including a schematic representation to convey the overall information would be beneficial for the audience's understanding. A visual representation can effectively communicate complex concepts and enhance comprehension. Therefore, it is recommended that the authors consider including a schematic representation in the manuscript to present the information in a more effective and visually engaging manner.
9. Overall, this review has the potential to offer valuable insights into the utilization of macro and micronutrients for the treatment and prevention of RVO. By incorporating the suggested revisions and additions, the present review can be further improved, making it more accessible and providing valuable information to a broader readership.
Author Response
Thank you very much for reviewing our manuscript. We wish to express our appreciation to you for your useful comments, which have helped us to improve our manuscript. According to the suggestions, we have revised our manuscript. We hope that you find our responses satisfactory.
Response to reviewer:
Reviewer wrote:
- Indeed, the title and abstract may create the impression that the manuscript is a research article rather than a review. To better reflect the nature of the paper, it would be advisable to modify the title and abstract to clearly indicate that it is a review article. For example: Title should be "A Comprehensive Review on the Prevention and Treatment of Retinal Vein Occlusion: The Role of Diet and Lifestyle"or "Prevention and Treatment of Retinal Vein Occlusion: The Role of Diet and Lifestyle – A Review".
Response: Thank you for your suggestion. Indeed, the title could mislead the reader. Corrected the title. Now It is "Prevention and Treatment of Retinal Vein Occlusion: The Role of Diet – A Review".
Reviewer wrote:
- The last sentence of the abstract, "The aim of this study is to check whether macro and micronutrients can affect retinal vein occlusion," does not seem suitable for a review paper. It may imply that the current manuscript is a research article. It would be better to rephrase it as follows: "The aim of this review is to investigate the potential impact of macro and micronutrients on retinal vein occlusion".
Response: Thank you for your comments. As with the first question, it also sounded here like a research article. We changed this statement. Now it sounds as you suggested : "The aim of this review is to investigate the potential impact of macro and micronutrients on retinal vein occlusion".
Reviewer wrote:
- The manuscript predominantly focuses on dietary habits for the prevention and treatment of RVO, with limited/no information provided about lifestyle changes. However, the title of the manuscript mentions both diet and lifestyle.
Response: Thank you for you remark. We changed the title. Now it is "Prevention and Treatment of Retinal Vein Occlusion: The Role of Diet – A Review".
Reviewer wrote:
- It would be beneficial to the audience if authors briefly discuss the range of treatment strategies currently available for RVO, beyond dietary and lifestyle adjustments, like anti-VEGF therapy, corticosteroids, laser therapy, intravitreal implants, retinal surgery, and more.
Response: Thank you very much for your suggestion. As we were advised, we extended treatment section.
The treatment of complications associated with RVO consists primarily of intravitreal administration of anti-VEGF preparations in the event of macular edema and photocoagulation of the entire retina in cases of iris neovascularization [7,8]. It has been proven that during RVO, vascular endothelial growth factor (VEGF) is the main cytokine inducing ischemia and neovascularization, therefore intravitreal anti-VEGF in the event of macular edema following retinal vein occlusion is the first line of treatment [1]. Aflibercept (EYLEA) is a fusion protein that binds VEGF-A, VEGF-B and placental growth factor (PIGF) with a greater affinity than the body’s native receptors. Ranibizumab (Lucentis) is a recombinant humanized IgG1 monoclonal antibody fragment that binds to and inhibits only VEGF-A. Bevacizumab (Avastin) is a humanized anti- body that binds all subtypes of VEGF-A. The recommended dose for aflibercept is 2 mg (0.05 mL), for bevacizumab it is 1.25 mg (0.05mL) and for ranibizumab it is 0.5 mg (0.05 mL) administered by intravitreal injection once every 4 weeks. After the first several injections, some patients continue monthly treatment, some patient are treated at increasing intervals and some patients are checked monthly and treated as needed. Faricimab (Vabysmo) is a promising bispecific drug targeting VEGF-A and the Ang-Tie/pathway [9]. It is a combined-mechanism medication with simultaneous and independent binding on both VEGF-A and angiopoietin-2 (Ang-2). It is believed to have a more lasting effect than previous anti-VEGF drugs in clinical trials. The FDA approved Vabysmo for the treatment of diabetic macular edema and neovascular age-related macular degeneration on January 2022. Another option to deliver effective anti-VEGF doses over a longer period of time is to use a slow-release intraocular device, such as a PDS device [9]. Patients who do not respond to anti-VEGF preparations are recommended to implement intravitreal steroid injection - triamcinolone or dexamethasone implant (DEX, Ozurdex) which reduce pathologically increased capillary permeability and inhibit the expression of cytokines and chemokines. Another method of treating macular edema is focal laser or grid laser which has now lost its importance because of intravitreal drug injections. In patients with ischemic RVO, panretinal laser photocoagulation (PRP) is recommended for treatment of secondary neovascular complications. Use of systemic recombinant tissue plasminogen activator (rtPA), radial optic neurotomy, chorioretinal anastomosis and arteriovenous sheathotomy are extremely rarely used methods due to possible complications such as sudden hemorrhage, visual field defects or retinal detachment [1]. Pars plana vitrectomy is considered in the presence of attached posterior hyaloids accompanied by persistent macular edema in CRVO. The recommended treatment methods are, unfortunately, associated with regular, lengthy visits to ophthalmological treatment facilities, which for people who are professionally active means exclusion from the labor market, and for the elderly and dependent people, family involvement in the treatment process.
[1] Ho, M.; Liu, D.T.L.; Lam, D.S.C.; Jonas, J.B. RETINAL VEIN OCCLUSIONS, FROM BASICS TO THE LATEST TREATMENT. Retina 2016, 36, 432-448, doi:10.1097/IAE.0000000000000843.
[9] Ghanchi, F.; Bourne, R.R.A.; Downes, S.M.; Gale, R.M.; Rennie, C.A.; Tapply, I.; Sobha Sivaprasad. An Update on Long-Acting Therapies in Chronic Sight-Threatening Eye Diseases of the Posterior Segment: AMD, DMO, RVO, Uveitis and Glaucoma. Eye (Lond) 2022, 36, 1154–1167, doi:10.1038/s41433-021-01766-w.
Reviewer wrote:
- Several genes have been implicated in the pathogenesis of RVO. It would be informative for the authors to explore whether the expression of these genes is influenced by the macro and micronutrients discussed in the manuscript. To enhance comprehension of the effect of nutrients on RVO pathogenesis, the authors may consider presenting a table or schematic representation that lists the genes known to be affected.
Response: Thank you for drawing attention to such an important issue. I will try to briefly describe the molecular relationships occurring in RVO. Some of these micronutrients have proven effects on these signaling pathways.
There is ample evidence that cytokines and chemokines contained in the vitreous body are correlated with the occurrence of RVO, especially the interleukin family, VEGF, MMP, LPA-ATX and PDGF [4]. The interleukin family is pro-inflammatory, causing ischemia and macular edema secondary to RVO, and the most important in this disease include: IL-6, IL-8, IL-17 and IL-18, which trigger STAT3, MAPK, NF-κB, VEGF pathways and provokes ROS. VEGF inhibits occludin by damaging the basement membrane of endothelial cells and activates MMP-9 to destroy blood-retinal barriers (BRB) and induces ICAM-1 causing leukocyte stasis. VEGF also works by activating the NOX1 and NOX4 proteins, which are dominant in ROS production in RVO. MMP-2 and MMP-7 are involved in the migration of vascular endothelial cells. The LPA-ATX signaling pathway may mediate inflammation in RVO as it activates IL-6, IL-8, VEGF and MMP-9. PDGF-A potentiates VEGF to induce neovascularization. - this paragraph will be added in the introduction
It has been confirmed that vitamin D has the potential for an anti-inflammatory response mainly by inhibiting the nuclear factor kapa B (NF-κB) signaling pathway and inhibiting pro-inflammatory factors [1]. In addition, it was found that vitamin D modulates apoptosis and reduces the expression of VEGF, inhibiting angiogenesis [2] . Vitamin D counteracts oxidative stress and reduces inflammation by reducing the expression of the protein interleukin 1, interleukin 8 and TNF-α. The listed anti-inflammatory, antioxidant and anti-angiogenic effects of vitamin D confirm its possible effect on the occurrence of RVO in conditions of its deficiency. - this paragraph will be added in the vitamin D section
- Caban, M.; Lewandowska, U. Vitamin D, the Vitamin D Receptor, Calcitriol Analogues and Their Link with Ocular Diseases.Nutrients 2022, 14, 2353, doi:10.3390/nu14112353.
- El-Sharkawy, A.; Malki, A. Vitamin D Signaling in Inflammation and Cancer: Molecular Mechanisms and Therapeutic Implications.Molecules 2020, 25, 3219, doi:10.3390/molecules25143219.
Vitamin A and C have proven abilities to capture reactive oxygen species and inhibit the activation of NF-κB, a transcription factor that promotes the expression of inflammation-inducing genes [1]. - - this paragraph will be added in the vitamin A, C section
- Conner, E. M.; Grisham, M. B. Inflammation, Free Radicals, and Antioxidants.Nutrition 1996, 12, 274–277, doi:10.1016/s0899-9007(96)00000-8.
Reviewer wrote:
- It would be valuable to provide the rationale for selecting the specific list of micronutrients discussed in this manuscript. Including this information at the appropriate place within the manuscript would enhance the understanding of why these particular micronutrients were chosen for discussion.
Response: Thank you for your suggestion. We do not hide that the issue we are considering is quite narrow and the number of available researches is small.
Pubmed was searched with no year limitations. The following keywords have been used: "diet", "vitamin" and "nutrients". Each of these keywords was used in combination with expression "retinal vein occlusion". Further searches were conducted combining the stated keywords with epidemiology, mechanisms, pathogenesis, neovascularization, cytokines, chemokines , plant-based diets, mediterranean diet and treatment. After reviewing the available literature, we have included relevant information on dietary habits and its impact on retinal vein occlusion. - these few sentences have been added at the end of section "introduction’’
Reviewer wrote:
- The content in the subheading "Covid-19 pandemic and lifestyle" appears to be unrelated to the context of the manuscript. It would be advisable to either remove or revise this subheading to ensure its alignment with the overall focus and scope of the manuscript.
Response: Thank you for your idea. With this short paragraph, we wanted to point out that the epidemic situation in the world also affects our eating habits. From our perspective, it is extremely important to highlight the impact of Covid on the increase in the incidence of thrombotic events in the eyes and all the more to emphasize the importance of proper nutrition in a pandemic situation. Nevertheless, we most certainly agree that the title does not fully correspond to the issue under discussion. We would like to replace it with "Covid-19 pandemic and dietary habits"
Reviewer wrote:
- Including a schematic representation to convey the overall information would be beneficial for the audience's understanding. A visual representation can effectively communicate complex concepts and enhance comprehension. Therefore, it is recommended that the authors consider including a schematic representation in the manuscript to present the information in a more effective and visually engaging manner.
Response: Thank you for the advice. We are in the process of creating such a graphic. It will most likely be included in the work at the next stage.
Reviewer wrote:
- Overall, this review has the potential to offer valuable insights into the utilization of macro and micronutrients for the treatment and prevention of RVO. By incorporating the suggested revisions and additions, the present review can be further improved, making it more accessible and providing valuable information to a broader readership.
Response: Thank you very much for the review. We are very pleased that manuscrypt received positive comments.

Round 2
Reviewer 3 Report
The authors have effectively addressed all the comments and suggestions provided, resulting in an enhanced manuscript that is clearer and more accessible, delivering valuable information to the readers. Based on these improvements, I recommend accepting the manuscript in its current format.
Author Response
Thank you very much for reviewing our manuscript. We wish to express our
appreciation to you for your useful comments, which have helped us to improve our manuscript.